# Workaholism and the Enactment of Bullying Behavior at Work: A Prospective Analysis

**DOI:** 10.3390/ijerph19042399

**Published:** 2022-02-19

**Authors:** Cristian Balducci, Luca Menghini, Paul M. Conway, Hermann Burr, Sara Zaniboni

**Affiliations:** 1Department of Psychology, University of Bologna, 40127 Bologna, Italy; luca.menghini3@unibo.it (L.M.); sara.zaniboni4@unibo.it (S.Z.); 2Department of Psychology, University of Copenhagen, 1353 Copenhagen, Denmark; paul.conway@psy.ku.dk; 3Work and Health Section, Federal Institute for Occupational Safety and Health (BAuA), 10317 Berlin, Germany; burr.hermann@baua.bund.de; 4Department of Management, Technology, and Economics, ETH Zürich, 8092 Zurich, Switzerland

**Keywords:** workaholism, workplace bullying behavior, cross-lagged relationship, negative affect, cross-lagged mediation

## Abstract

Despite the fact that workaholism and workplace aggressive behavior share many correlates, such as neuroticism, hostility, and negative affectivity, little is known about their relationship, with most evidence on both phenomena coming from cross-sectional studies. In the present study, we contributed to a better understanding of the antecedents of enacted workplace bullying behavior (i.e., perpetration of bullying), and the potential interpersonal implications of workaholism, by investigating their cross-lagged relationship. Data from a two-wave one-year panel study conducted with 235 employees in a national healthcare service organization showed substantial cross-sectional and cross-lagged positive relationships between workaholism and enacted workplace bullying. Whereas Time 1 workaholism was a significant predictor of Time 2 enacted workplace bullying, reversed causation was not supported. To shed light on the role of a potential mechanism explaining the link between workaholism and enactment of bullying, we examined whether job-related negative affect (e.g., anger) mediated their longitudinal relationship. However, whereas increased negative affect from T1 to T2 was positively associated with T2 enacted workplace bullying, the relationship between T1 workaholism and increased job-related negative affect was not significant, contrary to the hypothesized mediation. Taken together, our findings suggest that workaholism may be an important antecedent of enacted workplace bullying. Study limitations and future perspectives are discussed.

## 1. Introduction

In recent years, clinical, organizational, and occupational health scholars have paid increasing attention to the phenomenon of workaholism—an individual characteristic that manifests itself mainly through working very long hours, well beyond what is reasonably expected from a worker [1,2,3]. Underlying such a behavioral tendency, however, there is a persistent psychological dysfunction. Specifically, there is now a substantial degree of consensus that workaholism refers to a behavioral addiction, involving the feeling of being overly concerned about work and being urged by a strong, persistent, and uncontrollable inner drive, resulting in the need to spend exaggerated energy and effort in work, to the extent that private relationships, spare-time activities, and health may suffer [2,4,5,6]. Although workaholism has been also associated with positive attributes such as very high work effort [7], job satisfaction, and eustress [8,9], findings have consistently supported a negative view of workaholism, documenting its relationships with decreased mental health [10], and higher levels of negative emotions and anxiety [11,12], burnout [13], elevated blood pressure [14], and sickness absence [15]. Additionally, the notion that workaholics may perform better than their colleagues has been questioned, since the former have perfectionistic tendencies, including rigidity, inflexibility, and difficulties in delegating tasks [16,17], which are not instrumental to job performance. Indeed, a recent study found that workaholism was not associated with a supervisor-related measure of job performance [11], suggesting that there may be little advantage of being a workaholic for both individuals and organizations.

In the present study, we further contribute to the understanding of the drawbacks of workaholism by examining its interpersonal correlates (i.e., aggressive behavior at work), a topic that has received very little attention so far. Specifically, we focus on the enactment of workplace bullying behavior—a peculiar form of workplace aggression that, according to a common definition, means harassing, offending, or socially excluding someone at work, or negatively affecting someone’s work [18,19]. Workplace bullying has detrimental consequences for its victims and the organizations they work for, including anxiety, depression, sickness absence, intention to leave, early retirement from work, and, in extreme cases, suicidal ideation and attempts [20,21,22,23]. Workplace bullying is conceived as an extreme form of counterproductive work behavior (CWB) [24,25]. Thus, by investigating the relationship between workaholism and the enactment of bullying behavior, we also further contribute to the elucidation of the performance-related (i.e., organizational) implications of workaholism. Additionally, since workaholic tendencies are prevalent among managers and, more generally, among individuals with responsibility positions in organizations [26,27], evidence supporting a link between workaholism and the enactment of bullying would imply that workaholics have poor fit with managerial positions since they may actively contribute to create toxic workplaces.

### Theoretical Framework and Hypotheses Development

Workplace bullying has been mainly investigated from the perspective of the victim [16] and by frequently focusing on poor work environmental conditions as antecedents, such as workload, role conflict and ambiguity, and organizational change [28,29,30,31]. However, a number of studies have taken the perspective of the perpetrator (see [23,24]), suggesting that personal characteristics, such as neuroticism, hostile personality, and low emotional stability/self-regulatory capacity, may play a role as antecedents of the enactment of bullying. Such aspects have been considered as part of the so called “hot temperament” trait [32], which characterizes individuals who tend to handle stress by reacting aggressively. Indeed, research has shown that hot temperament is a precursor of workplace aggression (e.g., [33]).

Similarly, workaholism has been found in association with both neuroticism [12] and negative affectivity (see [3]), and, by definition, workaholics have low self-regulatory capacities, being unable to regulate their relationship with work despite the negative health-related consequences they may experience. Thus, workaholism shares a number of aspects with the hot temperament trait. In addition, previous research shows strong correlations between workaholism and both type A behavior and overcommitment [3,34], which reflect personal dispositions involving the frequent experience of negative work-related affective states, such as anger and hostility, which are two important cognitive components of aggressive behavior [35]. Workaholism is also associated with lower levels of prosocial workplace behavior [36] and with venting one’s negative emotions as a coping strategy [37]. Although discharging negative emotions may be beneficial for the individual (see [38]), it may come with significant interpersonal costs if it takes the form of inappropriate behavior. Overall, the above suggests that workaholism may be a significant antecedent for the perpetration of negative social behavior at work.

Theoretically, the link between workaholism and the enactment of bullying may be understood through the lenses of the stressor–emotion model of CWB [39], which builds upon the widely known frustration–aggression hypothesis [40]. According to this model, the experience of stress at work goes hand in hand with tension and frustration, and, more generally, negative affect, with the latter acting as the proximal trigger of aggressive behavior. Furthermore, the model postulates that work-related stress may be generated by both situational factors (i.e., working conditions) and predisposing factors related to individual differences. The latter, by acting as vulnerability conditions, predispose the individual to report frequently stress and frustration. Very similar ideas have been proposed by the General Affective Aggression Model [35], according to which personal factors such as negative affectivity, Type A behavior, and low self-esteem may lead to aggression by fueling the experience of critical internal states (hostile cognitions, physiological arousal, and negative affect).

By experiencing a strong, uncontrollable, internal drive to work excessively, and by investing a disproportionate amount of time and effort at work, workaholics frequently experience high levels of stress and frustration. This has been corroborated by studies adopting either a longitudinal design or within-person methods. For example, workaholism has been found to influence mental distress symptoms as reported one year later, controlling for the baseline level of mental distress [10]. Similarly, workaholism has been found to fuel daily job-related negative affect, over and above working conditions (workload) and other personal factors (neuroticism) [12]. Thus, in line with the stressor–emotion model of CWB, the enactment of aggressive behavior in the form of bullying may be a way for workaholics to deal with such high levels of stress experienced at work.

In a previous study, workaholism and enacted aggressive behavior were found to be significantly associated [41], with their relationship being partially mediated by negative job-related affect, which is in line with what may be predicted by the stressor–emotion model of CWB. However, this study was cross-sectional in nature, resulting in a high vulnerability to common method bias and limited inference about the causal nature of the relationship. In general, scholars in the fields of both workaholism [5] and workplace bullying [18] have repeatedly claimed that much of what we know about the two phenomena is based on cross-sectional evidence, and have called for more longitudinal research. Thus, to advance our understanding on the antecedents of workplace bullying and the potential implications of workaholism, we conducted a two-wave study testing the following hypothesis:

**Hypothesis** **(H1).**
*Workaholism at Time 1 (T1) will have a positive impact on the enactment of workplace bullying as reported at Time 2 (T2), over and above the baseline level of enacted bullying behavior.*


Additionally, building on previous evidence [37] and theoretical elaborations clearly suggesting that negative affect is a typical experience among workaholics [12], and an important proximal antecedent of workplace aggressive behavior [42], we sought to examine the potential mediation of negative job-related affect (e.g., anger) by testing the following hypothesis:

**Hypothesis** **(H2).**
*The relationship between workaholism at T1 and the enactment of bullying behavior at T2 will be mediated by job-related negative affect.*


## 2. Materials and Methods

### 2.1. Participants and Procedure

Data were collected in a two-wave, one-year lagged study in a national healthcare service organization in Northern Italy. The investigation was part of a workplace psychosocial risk assessment conducted in accomplishment of the Italian health and safety law (Legislative Decree 81/2008). The survey involved the departments of the organization that scored highest on a number of potential indicators of work-related stress (e.g., accidents at work, turnover, sickness absences) (see [43]). Such indicators were first collected from organizational databases by the HR department and then analyzed at the aggregated (i.e., department) level. This is a standard procedure based on the national guidelines for psychosocial risk assessment (see [44]), which requires a preliminary phase focusing on objective indicators of work-related stress, and a successive in-depth phase (which is the focus of the present study) consisting of employee surveys on psychosocial risk factors. The researchers were only involved in the latter phase. Data collection was undertaken by means of an anonymous paper-and-pencil questionnaire, which was administered during working time. Specifically, different sessions of data collection were organized with the help of “project development facilitators” (i.e., workers of the identified departments, such as safety representatives) that were present during the data collection, and were available to deal with requests for clarification from the participants. All the employees of the selected departments were invited to participate. Ad hoc boxes were placed in the various departments/wards for returning the completed questionnaires at the end of the data collection sessions. Questionnaires were then delivered to the researchers, who proceeded to build the dataset used for both the organizational risk assessment required and the present study.

The questionnaire was completed by 574 employees at T1, with a response rate of 75.4% across the various departments (T1 data were used in a previous cross-sectional study exploring the workaholism-bullying relationship [41]). At T2, the questionnaire was completed by 508 employees, with a response rate of 65%. T1 and T2 questionnaires were matched using anonymous codes built using factual personal information. This procedure ensured responses were anonymous while allowing the linking of baseline (T1) and follow-up (T2) data. Follow-up data were available for 235 employees (40.9% of those completing the T1 questionnaire). Participants were predominantly females (86.3%), and mainly aged 30–39 (37%) and 40–49 years (38%). The majority of participants were nurses (73%) and administrative staff (18%). Most participants had a permanent job contract (97%) and a job tenure equal to or higher than five years (83%).

### 2.2. Measures

Workaholism was measured with the Dutch Work Addiction Scale [45], which consists of 10 items that investigate a number of feelings related to the respondent’s work and reflect the two hypothesized components of workaholism, namely, working compulsively (e.g., “I feel that there’s something inside me that drives me to work hard”) and working excessively (e.g., “I stay busy and keep many irons in the fire”). Responses were given on a four-point Likert scale, ranging from 1 (“Never or almost never”) to 4 (“Almost always or always”). The Dutch Work Addiction Scale has been adapted to Italian, showing good internal consistency and test–retest stability. It also showed significant negative correlations with measures of well-being and positive correlations with measures of poor mental health such as anxiety and depressive symptoms [46]. For the present study, we used the overall scale score, which showed good internal consistency at both T1 (Cronbach’s α = 0.82, 95% CI [0.78, 0.85]) and T2 (α = 0.81, 95% CI [0.77, 0.84]).

Job-related negative affect was assessed by four items taken from a shortened version (see [47]) of the Job-related Affective Well-being Scale [48], which has already been used in previous research with Italian employees (e.g., [10,41,46]). The scale investigates the frequency with which the participant has experienced each of a number of affective states in relation to his or her work across the previous 30 days, with responses given on a five-point Likert scale from 1 (“Never”) to 5 (“Very often”). The affective states assessed in the present study were anger, disgust, pessimism, and discouragement. The internal consistency of the derived scale was α = 0.79, 95% CI [0.75, 0.83] and α = 0.77, 95% CI [0.72, 0.81], respectively, at T1 and T2.

Enactment of workplace bullying behavior was measured by adapting nine items of the Italian version [49] of the Short Negative Acts Questionnaire [50,51]. This scale assesses the experience of bullying from the perspective of the victim (e.g., ‘‘Someone at work has persistently criticized your work and effort’’). To investigate the enactment of bullying (the dependent variable in the present study), i.e., to take the perspective of the perpetrator, the items of the scale were rephrased in an active form (e.g., ‘‘You have persistently criticized the work and effort of someone at work’’). Responses varied from 1 (‘‘Never’’) to 5 (‘‘Daily’’), with respondents asked to consider the last six months. Cronbach’s α for this scale was slightly below the commonly accepted threshold of 0.70, (i.e., α = 0.67 95% CI [0.60, 0.73] at T1, and α = 0.65 95% CI [0.58, 0.71] at T2). However, it has been argued that for less clearly-delimited psychological phenomena (of which aggressive behavior can be considered an example), measurement scales that attain an alpha of 0.60 to 0.70 can be regarded as acceptable (see [52]). These items have already been used in previous research (e.g., [53,54]) to study bullying behavior from a perpetrator’s perspective.

### 2.3. Data Analysis

We conducted a cross-lagged path analysis using the Mplus 8.4 software (Muthén and Muthén, Los Angeles, CA, USA). We opted for path analysis rather than relying on structural equation modelling (SEM) because SEM is best suited for large sample sizes (>300) [55], which was not our case. All the information available from the 235 participants that responded at both T1 and T2 was used to fit the models described below (i.e., no listwise deletion was implemented). To test H1, we first specified a baseline model (Model 1) including the cross-sectional covariation between workaholism and enactment of workplace bullying behavior at both T1 and T2, and the auto-regressive paths (e.g., from T1 workaholism to T2 workaholism). Model 1 also included gender as a covariate affecting T2 enactment of bullying behavior, since research has often shown that males tend to be more aggressive than females (e.g., [32]). Then, in a second model (Model 2), we included the cross-lagged path from T1 workaholism to T2 enactment of bullying behavior, and checked whether the path was statistically significant and whether the resulting model provided a significantly better fit than the baseline model. In a third model alternative to Model 2 (Model 3), we also assessed reversed causation (i.e., from T1 enactment of workplace bullying behavior to T2 workaholism).

To test H2, we ran a simple linear regression analysis with T2 job-related negative affect regressed on T1 job-related negative affect, and used the obtained residuals to represent change in job-related negative affect from T1 to T2 (i.e., Δ Job-related negative affect) (e.g., see [56]). This variable was then added to Model 2 as a mediator between T1 workaholism and T2 bullying. Specifically, a baseline model (Model 4), corresponding to Model 2 plus the inclusion of an uncorrelated Δ Job-related negative affect variable, was compared to the hypothesized model (Model 5), which also included the paths from T1 workaholism to Δ Job-related negative affect, and from Δ Job-related negative affect to T2 enactment of workplace bullying behavior (see graphical representations in the Results section).

Models 1–5 were specified using the Robust Maximum Likelihood estimator, and their goodness of fit was assessed and compared in terms of the χ^2^ statistic, comparative fit index (CFI), Tucker−Lewis index (TLI), root-mean-square error of approximation (RMSEA), and standardized root-mean-square residual (SRMR). The difference in χ^2^ between competing models was computed using the Satorra and Bentler scaled χ^2^ difference test (See: http://www.statmodel.com/chidiff.shtml (accessed on 14 February 2022)). As standard practice, TLI and CFI values greater than 0.90 and RMSEA and SRMR values lower than 0.08 were deemed acceptable (see [57]).

## 3. Results

We first performed an attrition analysis to test whether drop-out at T2 was related to the main study variables. Specifically, we conducted a logistic regression analysis on the 574 participants in the T1 survey, in which drop-out (0 = participation in follow-up; 1 = drop out) was predicted by the main study variables (i.e., T1 workaholism, T1 enactment of bullying behavior, and T1 job-related negative affect). Results revealed that T1 workaholism was a risk factor for drop out (Odds Ratio (OR) = 1.56, *p* < 0.05), whereas none of the other variables were related to drop-out. Although these results may have determined a range restriction in workaholism with potential implications for the main analyses, selection bias is thought to significantly impact the results if drop-out from the study is related to both the predictor(s) and the outcome variable [58], which was not the case in the present study.

Table 1 reports means, standard deviations, and intercorrelations between the study variables. Both cross-sectional and longitudinal correlations between workaholism and enactment of bullying behavior were positive and in the low-to-moderate range (0.15 ≤ *r* ≤ 0.28). Additionally, job-related negative affect at T1 and T2 was also weakly-to-moderately related to both workaholism and enactment of bullying at both time points.

To test H1, stating that workaholism has a cross-lagged effect on enacted workplace bullying, we first estimated the baseline model (Model 1), which fitted the data well (χ^2^(5) = 4.77, CFI = 1.00; TLI = 1.00; RMSEA = 0.00 [95% CI =0.000–0.010]; SRMR = 0.029). We then fitted a second model (Model 2) which, in addition to the paths included in Model 1, included the cross-lagged path between T1 workaholism and T2 enactment of bullying behavior. Model 2 (see Figure 1) also fitted the data well (χ^2^(4) = 1.07, CFI = 1.00; TLI = 1.00; RMSEA = 0.00 [0.000–0.043]; SRMR = 0.014). The Satorra and Bentler [59] scaled χ^2^ difference test comparing Model 1 with Model 2 was significant (χ^2^(1) = 3.84, *p* = 0.05). Importantly, the cross-lagged path between T1 workaholism to T2 enactment of bullying behavior in Model 2 was significant, even if small in size (std. path = 0.10, *t*(1) = 2.03, *p* < 0.05). Overall, these results lend support to H1. In a further model (Model 3), we tested reversed causation, that is, the effect of T1 enactment of bullying behavior on T2 workaholism. Although Model 3 fitted the data well (χ^2^(4) = 4.47, CFI = 0.998; TLI = 0.996; RMSEA = 0.022 [CI = 0.000–0.104]; SRMR = 0.030), the estimated relationship between workplace bullying behavior and T2 workaholism was negligible and not significant, indicating that reversed causation did not hold (see Figure 1). In addition, removing gender from Models 2 and 3 did not change the statistical significance of the T1 workaholism–T2 enactment of workplace bullying relationship.

Subsequently, we tested H2, stating that job-related negative affect mediated the relationship between T1 workaholism and T2 enactment of bullying behavior. First, we estimated the baseline model for testing mediation (Model 4), which provided the following fit to the data (χ^2^(10) = 28.51, CFI = 0.923; TLI = 0.908; RMSEA = 0.089 [0.052–0.129]; SRMR = 0.074). We then fitted a further model (Model 5) which, in addition to the paths included in Model 4, included the path from T1 workaholism to Δ Job-related negative affect, and the path from Δ Job-related negative affect to T2 enactment of bullying. Model 5 fitted the data satisfactorily (χ^2^(8) = 19.29, CFI = 0.953; TLI = 0.930; RMSEA = 0.078 [0.033–0.123]; SRMR = 0.044) and, according to the Satorra and Bentler scaled χ^2^ difference test, significantly better than Model 4 (χ^2^(2) = 8.26, *p* = 0.002). Nevertheless, as shown in Figure 2, although the path from Δ Job-related negative affect to T2 enactment of workplace bullying behavior was statistically significant (std. path = 0.13, *t*(1) = 2.34, *p* < 0.05), the path from T1 workaholism to Δ Job-related negative affect was not (std. path = 0.13, *t*(1) = 1.78, *p* = 0.08), implying that the preconditions for testing mediated relationships were not met. Therefore, these results did not support H2.

## 4. Discussion

Workplace bullying behavior is a form of CWB involving harassment, offenses, and social exclusion [18,24,25]. The enactment of such negative acts at work, i.e., being a perpetrator, has been associated with negative affectivity and with neurotic, emotionally dysregulated, competitive, and hostile personality profiles [32,60]. Interestingly, workaholics can be characterized by similar personality traits [3,12,34], suggesting their propensity to enact bullying behavior at work. However, the relationship between workaholism and the enactment of bullying behavior at work has been rarely investigated. In the present study, we aimed to advance our knowledge of the relationship between workaholism and workplace bullying by investigating the longitudinal association between the two constructs. Building on the theoretical frameworks of the stressor–emotion model of CWB [39] and the general affective aggression model [35], we hypothesized that workaholism may fuel an individual’s high levels of stress along with their associated critical internal states (i.e., frustration and negative affective experiences), which, in turn, may lead to the enactment of bullying behavior.

In line with our first hypothesis (H1), and corroborating a previous cross-sectional study [41], we found that workaholism predicted higher levels of enactment of bullying behavior at work as reported one year later, after adjusting for gender and the stability of both variables. By working compulsively and excessively, workaholic individuals may experience higher levels of work-related stress, which is likely to explain their higher levels of enactment of bullying behavior. This result contributes to the characterization of the performance-related and interpersonal implications of workaholism, which have been rarely considered so far, thus expanding our knowledge of the nomological network of workaholism. The fact that previous studies failed to support a negative relationship between workaholism and employee performance (e.g., [11]) may be explained by the lack of distinction between the different facets of job performance (i.e., in role, extra role, and CWB) (see [25]). On the contrary, by focusing on the enactment of bullying behavior, we were able to show that workaholism does not only have individual costs in terms of the health and well-being of the affected worker [11,12,13,14,15], but may also have interpersonal- and performance-related—and thus organizational—costs. Given the dire consequences that exposure to bullying behavior may have on targets, bystanders, and work groups (e.g., turnover, sickness absence, and loss of motivation and engagement) [19], such costs may be substantial [61].

Contrary to the second hypothesis, however, our study did not support previous cross-sectional evidence [41] about the mediating role of job-related negative affect in the relationship between workaholism and the enactment of bullying behavior. It is possible that the one-year time lag adopted in the present study was inadequate to capture the assumed intervening affective process. Indeed, negative job-related affective experiences may represent short-term stress reactions fueled by workaholism, so the impact of workaholism on such reactions may be better identified over shorter time intervals [12], whereas longer time lags, such as the one adopted in the present study, may be more suitable to investigate the impact of workaholism on long-term stress reactions such as mental health symptoms (e.g., burnout, depression), which were not considered in the present study. Thus, future research should implement longitudinal designs with shorter time lags, or even diary studies focusing on daily or weekly job-related negative affect and enactment of bullying behavior. In addition, because a central component of workaholism is working compulsively—which, by definition, is an anxiety-related phenomenon [1]—it may be the case that workaholism already houses the affective ingredients that trigger aggressive behavior. Therefore, including further affective components may not add substantially as far as the relationship between workaholism and the enactment of bullying is concerned.

Our study also contributes to advancing our understanding of the possible individual antecedents of aggressive behavior at work, by suggesting that workplace bullying may be more likely enacted by individuals with workaholic tendencies. Previous studies on bullying have highlighted the important role of personal characteristics as antecedents of the phenomenon, such as lack of self-regulative capacities [60], low agreeableness and conscientiousness, and higher neuroticism. With the present study, we shed light on workaholism as a further personal disposition that may be implicated in the enactment of bullying behavior. Of note, workaholism may not be a trivial antecedent of bullying, since estimates suggest that workaholism is prevalent in the working population, ranging from 6.6% to 20.6% (see [62]). It would be interesting, in future research, to not only replicate the present findings among other types of workers with a higher prevalence of workaholism (e.g., managers [26]), but also to examine the potential interaction between workaholism and work environmental factors in the process leading to bullying behavior, which has been indicated as a priority in bullying research [19].

In terms of implications, the results of the present study suggest that organizations should avoid reinforcing workaholic tendencies in individuals, since such tendences may impair interpersonal relationships with colleagues, and with patients and customers. Preventing workaholic tendencies may translate, for example, into combating the widespread workplace culture that instigates overwork and competition [63], which is also highly prevalent in the health sector (see [62]) (i.e., the context of the present study). Another strategy could be the promotion of managerial styles and practices that are more family-friendly [64], and that actively encourage detachment and recovery [65]. It is increasingly acknowledged that working conditions may significantly impact personality, with changes occurring even in a relatively short period of time [66]. For example, it has been shown that high workload may contribute to the development or consolidation of workaholic tendencies [10]. Thus, by taking action against the factors that may reinforce workaholic tendencies, organizations can not only preserve the health and vitality of their employees, but also defuse a significant predictor of aggressive behavior and bullying.

A further implication of the present study concerns the fit of the workaholics for supervisory or managerial positions, considering that being a manager is the most important risk factor for workaholism (see [27]). First, workaholic managers, by acting as role models for their subordinates, may lead them to become workaholics themselves, especially if the latter have predisposing personal characteristics such as high perfectionism or obsessive-compulsive personality traits. This process of transmission of workaholic tendencies may contribute to create what has been called the “addictive organization” [67], which may have substantial human costs in terms of burnout and overwork-related disorders [1]. Additionally, by enacting bullying towards their immediate subordinate, workaholic managers may initiate “waves” of aggression that trickle down to lower levels of the organizational hierarchy through the mechanism of displaced aggression [33], whereby the target of bullying behavior becomes a perpetrator towards less-powerful others. If employees regularly interact with patients or customers, as is the case in the sample examined in the present study, aggression may also be directed towards them. Such processes may clearly lead to the development of an interpersonally toxic work environment. In summary, a workaholic leader may thwart the potential of employees to thrive at work and create significant costs for their organization.

The results of the present study should be interpreted in the light of some limitations. First, only self-report data were used, implying that the observed relationships may at least partially be explained by common method bias, although the longitudinal design is expected to reduce this type of bias (see [68]). Whereas the focus on the perpetrator perspective is appropriate for investigating the individual and personality-related correlates of workplace bullying behavior (e.g., [32,60]), self-reports of the enactment of aggressive behavior are likely biased by social desirability, implying that further studies employing alternative methods, such as supervisors and co-workers’ reports of workplace bullying, are needed for reaching more solid conclusions about the relationship between workaholism and bullying. Additionally, the adopted measure of bullying displayed a suboptimal internal consistency, meaning that measurement error for the construct may have been substantial, potentially affecting the obtained results. In additional analyses (available upon request from the first author) we found that one of the items of the measure (“I carried out practical jokes on someone I don’t get on with”) had a very low loading at one measurement occasion on the underlying factor (i.e., <0.10). However, discarding the item did not significantly increase the internal consistency of the measure (alpha was 0.68 and became 0.69). Future studies should pay attention to this problematic item when using the SNAQ as a measure of enacted bullying, and perhaps consider deleting or reformulating it. It should also be noted, however, that, according to some scholars [52], behavioral items of aggression such as the one adopted here may be considered as “formative” instead of “reflective” indicators, meaning that internal consistency and factor analysis may not be appropriate to assess their structure and reliability. Second, our longitudinal study only included two waves of measurement, providing a limited information on the cross-lagged relationship between the investigated constructs, especially on the mediating variables; study designs including three or more waves are needed to shed more light on the hypothesized mediating processes [69]. Third, our sample was relatively small, largely consisting of female nurses, and involved individuals employed in the same public organization, resulting in a limited generalizability to other organizational contexts (e.g., private sector), jobs, and sectors. Future studies should replicate our results on larger and job-specific samples. Additionally, future studies could focus on high-responsibility positions (e.g., managers, higher-level professionals, and self-employed) that typically display a higher prevalence of workaholism [26,27].

## 5. Conclusions

In conclusion, our study adds to the previous literature by showing that workaholism may be a potential antecedent of workplace aggressive behavior. The main implication of our results is that employers and practitioners have an additional reason for planning and conducting interventions aimed at preventing and reducing workaholism in the workforce, because it may not only have a negative impact on workaholics’ health and well-being, but also promote the enactment of CWB in the form of bullying, with negative consequences on employees and the organization overall.

## Figures and Tables

**Figure 1 ijerph-19-02399-f001:**
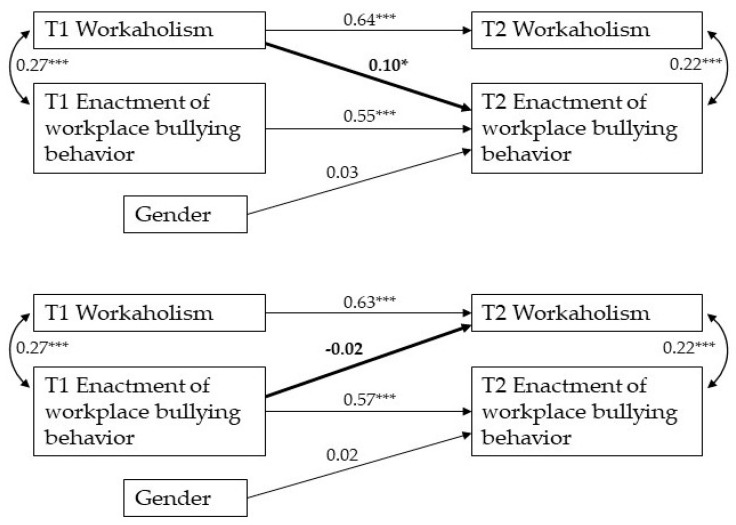
Path analysis in Model 2 (above) and Model 3 (below) of the cross-lagged relationships between workaholism and enactment of workplace bullying behavior. * *p* < 0.05, *** *p* < 0.001.

**Figure 2 ijerph-19-02399-f002:**
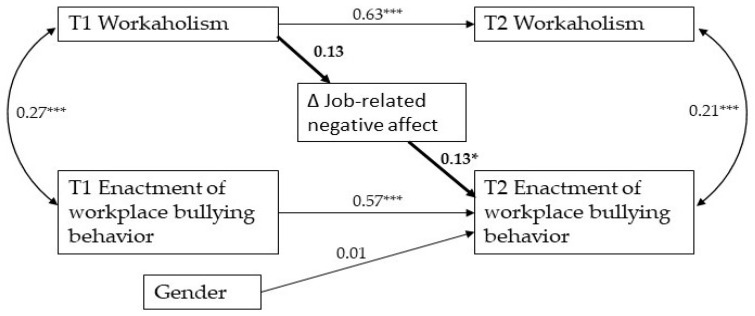
Path analysis in Model 4 of the cross-lagged relationships between workaholism, job-related negative affect, and enactment of workplace bullying behavior. * *p* < 0.05, *** *p* < 0.001.

**Table 1 ijerph-19-02399-t001:** Descriptive statistics and zero-order Pearson correlation coefficients between the study variables.

	n	Mean (SD)	Range	1	2	3	4	5
1. Workaholism T1	235	2.10 (0.54)	1.00–3.80					
2. Workaholism T2	234	2.12 (0.52)	1.00–3.78	**0.64 *****				
3. Enactment of bullying T1	235	1.31 (0.30)	1.00–2.89	0.27 ***	0.15 *			
4. Enactment of bullying T2	234	1.20 (0.31)	1.00–2.44	0.25 ***	0.28 ***	**0.57 *****		
5. Job-related negative affect T1	234	2.25 (0.83)	1.63–5.00	0.33 ***	0.23 ***	0.14 *	0.21 **	
6. Job-related negative affect T2	233	2.36 (0.79)	1.00–5.00	0.28 ***	0.33 ***	0.13	0.28 ***	**0.52 *****

Notes: *** *p* < 0.001; ** *p* < 0.01; * *p* < 0.05. SD = standard deviation. Bold types indicate auto-correlations over the two time points.

## Data Availability

The data are available upon request from the first author.

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
