# Peer review of "Workaholism and the Enactment of Bullying Behavior at Work: A Prospective Analysis"

_ijerph, 2022, doi:10.3390/ijerph19042399_

Round 1
Reviewer 1 Report
Thank you for the submission entitled “Workaholism and the Enactment of Bullying Behavior at Work: A Prospective Analysis.” This is an exciting study of essential phenomena, i.e., workaholism and bullying behaviors. The authors used cross-lagged and cross-sectional study designs to answer the research questions related to workaholism, workplace bullying behaviors, and negative affect. The study findings suggest that workaholism is a critical antecedent of enacted workplace bullying among healthcare professionals.
Although the paper seems to focus on an important topic, specific problems in this manuscript deserve attention. I present them here to make some ideas for strengthening the article.
The authors should focus more on addressing what we already know about the topic before bringing in a gap considering what the paper tries to fill in. This would make it clear to the reader why it is crucial to address the shortcomings in the literature. I believe that your contribution would be more significant if you present convincing arguments regarding why there is a need to investigate the underlying questions. What new insight is your study offering to readers? If you convince the reader of the necessity for your work to expand our current knowledge in the introduction section, you would significantly enhance your contribution.
Please add the theoretical perspective to explain the association and build arguments among study variables.
I suggest reviewing the past studies, identifying all the predictors and outcomes of workaholism and workplace bullying, and putting them in a table with relevant sources in the literature review to understand both the phenomenas better. You could introduce a pre-version of this table in the introduction in the form of a paragraph. With the help of this one, one can easily understand what we know and what we do not know.
Why job-related negative affect is best to explain the association between workaholism and workplace bullying. More justifications are required with the help of literature that explored the mediating role of job-related negative affect.
Can you describe how you recruited the sample? Participants and procedures should need to elaborate in more detail.
Please justify why MPlus software is best to investigate the study paths instead of other techniques like SEM.
The convergent and discriminant validity needs to be performed before testing the direct and indirect paths.
The discussion part is underdeveloped. In particular, the discussion needs to be more thorough and linked to the literature review.
The theoretical implications part is underdeveloped.
I am very disappointed that most of the references are very old. The more references that come from the past five years, the better. Therefore, I suggest that the authors need to incorporate recent and more context-related articles related to the phenomenon.
In general, I would like my recommendations to help the authors improve their work. I hope the authors will benefit from these suggestions and make the necessary amendments to improve the work for later submission.
Author Response
Please see attachment

Dear Editor and Reviewers,
thank you for giving us the possibility to submit a revised version of our manuscript (MS.) on workaholism and the enactment of bullying behavior. We have appreciated very much the constructive tone of both reviews. We respond below point-by-point to each Reviewer comment – see text in red colour. Addressing all the comments required quite much work. We hope that we have been able to strengthen our MS., making it suitabile for publication on IJERPH. However, if you have any remaining/additional comment and suggestion, we will be happy to address them.
Changes in the manuscript are highlighted with red and underlined font.
Best regards,
Cristian Balducci,
On behalf of all the authors.
Reviewer 1 comments
Although the paper seems to focus on an important topic, specific problems in this manuscript deserve attention. I present them here to make some ideas for strengthening the article.
Dear Reviewer,
first of all thank you very much for reading and commenting our study. Thank you also for being constructive and giving specific suggestions on how to strengthen our manuscript (MS.). We reply here to your fist block of comments regarding the Introduction of the MS.
Changes in the manuscript are highlighted with red and underlined font.
The authors should focus more on addressing what we already know about the topic before bringing in a gap considering what the paper tries to fill in. This would make it clear to the reader why it is crucial to address the shortcomings in the literature. I believe that your contribution would be more significant if you present convincing arguments regarding why there is a need to investigate the underlying questions. What new insight is your study offering to readers? If you convince the reader of the necessity for your work to expand our current knowledge in the introduction section, you would significantly enhance your contribution.
Please add the theoretical perspective to explain the association and build arguments among study variables.
I suggest reviewing the past studies, identifying all the predictors and outcomes of workaholism and workplace bullying, and putting them in a table with relevant sources in the literature review to understand both the phenomenas better. You could introduce a pre-version of this table in the introduction in the form of a paragraph. With the help of this one, one can easily understand what we know and what we do not know.
Why job-related negative affect is best to explain the association between workaholism and workplace bullying. More justifications are required with the help of literature that explored the mediating role of job-related negative affect.
Following your comments above, we have substantially extended the Introduction (pp. 1-6). We have now introduced a paragraph on the theoretical frameworks that can be used to explain the relationhip between workaholism and the enactment of bullying behavior. We opted for an established model - the stressor emotion model of counterproductive work behavior proposed by Spector and Fox (2005), which builds upon the frustration-aggression hypothesis as revisited by Berkowitz (e.g., 1989). Such a framework is particularly suited to shed light on the relationship between workaholism and enactment of bullying, since it postulates that aggression (i.e., bullying) may be a consequence of negative affective states (frustration and tension) experienced in relation to external stressors (e.g., negative working conditions) or personal dispositions (‘internal’ factors) fueling the experience of stress. We also mention an additional theoretical model developed in social psychology, namely the General Affective Aggression Model (Allen et al., 2018 – see reference list in the MS.), which makes predictions very similar to the stressor-emotion model, that is, aggression may be a consequence of individual differences that fuel critical internal states such as negative affect.
As we now explain by describing in more detail the results emerged from recent sound studies (e.g., the two studies by Balducci et al., 2021), workaholism may fuel work-related stress and expose the individual to frustration, tension, and negative affect, creating the necessary and sufficient internal experiences that trigger the enactment of bullying behavior. In this way, we have provided the requested theoretical rationale for our study, also explaining “why job-related negative affect is best to explain the association between workaholism and workplace bullying”. On the latter point, we have also included some recent sound studies indicating that workaholics indeed experience stress and negative job-related affect, making it plausible that negative affective experiences are a crucial mediator of the workaholism-enactment of bullying relationship.
In the opening section of the Introduction (p. 3), we also provided additional reasons, as per your request above, to investigate the workaholism-enactment of bullying relationship. We have now emphasised that workaholics often have responsibility positions, so if we demonstrate that workaholism is related to the enactment of bullying we will provide important evidence that workaholics may not be the best fit for managerial positions, since they contribute to create toxic workplaces.
We would avoid, however, to extend too much the Introduction with the addition of more text or tables regarding both workaholism and workplace bullying, which are well known phenomena and for which we have now inclued additional key references, also considering that the authors’ guidelines of IJERPH stress that the Intro should be brief and to the point: “The introduction should briefly place the study in a broad context and highlight why it is important”.
In sum, we believe that the revised version of our manuscript fits well with IJERPH guidelines while, at the same time, responding to the request of identifying the core results of the past literature and providing a more comprehensive theoretical rationale for our study.
-Can you describe how you recruited the sample? Participants and procedures should need to elaborate in more detail.
We have given additional information for the section “Participants and procedure” (p. 3), hoping that this will provide sufficient details on the recruitment procedures. Alternatively, please let us know which specific point is in need of further clarification.
-Please justify why MPlus software is best to investigate the study paths instead of other techniques like SEM.
We believe this comment raised from a misunderstanding of the difference between Mplus (i.e., a widely used data analysis software focused on SEM, similar to other softwares such as Lisrel or the R package lavaan) and SEM (i.e., a class of models that use the covariance matrix for estimating target model parameters, and which can be implemented with the Mplus software). Path analysis is just a specific type of SEM that uses observed variables only (i.e., aggregate scales scores, with no latent variables and factor loadings), and which is routinely used to conduct multivariate analyses, especially when the sample size is not sufficiently large to estimate all the parameters involved in SEM with latent variables. Thus, we did not find necessary to justify the use of Mplus to investigate the study path, since both the Mplus software and SEM with observed variables are accepted and widely used by the scientific community. However, to account for possible misunderstandings, we now referenced (p. 5) to Tabachnick and Fidell (2013). Using multivariate statistics, Boston: Pearson Education.
-The convergent and discriminant validity needs to be performed before testing the direct and indirect paths.
Some results are already evident from the correlation table (see Table 1), showing that, in all cases, the measures of the same construct at different times (T1 and T2) correlate more strongly than the measures of different constructs.
Additionally, we conducted two confirmatory factor analyses (estimator: robust RML), one at T1 and one at T2 to test for whether the three constructs under investigation (i.e, workaholism, enactment of bullying, and job-related negative affect) could be discriminated empirically. In these analyses, also to account for the relatively low sample size, each construct was operationalized by means of two parcels (i.e., subscales). For example, workaholism was operationalized by two 5-item parcels, one reflecting working excessively, and the other reflecting working compulsively. At T1, a 3-factor model fitted the data sufficiently well (χ2(6) = 13.49, CFI = .96; RMSEA = .073; SRMR = .030) and significantly better than a 1-factor model (Δχ2(3) = 76.44, p<.001). At T2 the results were similar, with the 3-factor model (χ2(6) = 5.72, CFI = 1.00; RMSEA = .000; SRMR = .027) fitting significantly better than the 1-factor model (Δχ2(3) = 119.15, p<.001).
Currently, we did not include such results in the Results section since the three considered measures are well validated and have been employed multiple times in previous research, and our study did not focus on construct validity. However, if you believe we should integrate the above results in the MS., we will be happy to do so.
-The discussion part is underdeveloped. In particular, the discussion needs to be more thorough and linked to the literature review.
-The theoretical implications part is underdeveloped.
As suggested, we substantially expanded the Discussion (pp. 8-9), particularly by better linking our results to the most relevant literature, and by better elaborating on the implications of our results. We also acknowledged additional limitations in reply to the Reviewer 2. In all, there are three additional paragraphs in the Discussion. We believe the current version of the Discussion is now well developed, providing the necessary information to interpret our results, to understand the study limitations, and to appreciate both the practical and theoretical implications of our study.
I am very disappointed that most of the references are very old. The more references that come from the past five years, the better. Therefore, I suggest that the authors need to incorporate recent and more context-related articles related to the phenomenon.
We have now added more recent references, also in response to the other comments of both reviewers. However, we highlight that thirty out of fifty-five references (i.e. the majority) reported in the first version of the MS. were published in 2016 or later, so we kindly disagree with your statement that “most references are very old”.
In general, I would like my recommendations to help the authors improve their work. I hope the authors will benefit from these suggestions and make the necessary amendments to improve the work for later submission.
Dear Reviewer, thank you again for your recommendations, and we hope that, by following your and Reviewer 2’s suggestions, we have been able to improve our MS. to the point of being accepted for publication on IJERPH.
Reviewer 2 comments
General: This article is quite original in its approach to studying workaholism, bullying (from the the perpetrator's perspective) and the role of anger. It's an overall very interesting article. The longitudinal perspective and the inclusion of anger as a potential mediator in the relationship between workaholism and bullying is a novelty and merits to be pursued in the future. However, the design of the study has some flaws. For instance, the effect found for H1 seems quite small, most probably due to measurement issues. Moreover, the sample represents an overly large proportion of females (86%), which may have biased the results. Also, the dropout rate seems problematic from T1 to T2 and may also explain why the relationship between workaholism and bullying is so small. Recommendations on how to improve future longitudinal studies to examine the relationships between workaholism, bullying at work and anger give some good indications. Several more suggestions are described for each section.
Dear Reviewer 2, thank you very much for appreciating our study and for your comments and suggestions on how to improve it. We have now acknowledged the limitations you report above in a more elaborated way (p. 10), also giving suggestions on how future studies could overcome (some of) them.
Abstract: It's not clear whether anger mediates the relationship or not. The sentence summarizing the effect found at T1 and T2 (lines 21-23) is not very clear.
Thank you for noting this. We agree with you and have now better clarified that part:
- 1
“To shed light on the role of a potential mechanism explaining the link between workaholism and enactment of bullying we examined whether job-related negative affect (e.g., anger) mediated their longitudinal relationship. However, whereas increased negative affect from T1 to T2 was positively associated with T2 enacted workplace bullying, the relationship between T1 workaholism and increased job-related negative affect was not significant, contrary to the hypothesized mediation.”
Introduction: The first two sentences are a little too broad and generic. The introduction may benefit from stronger argumentation and a few specific numbers perhaps demonstrating the changes at work (lines 30-33). Which Western countries do you include here?
We agree also on this. However, we decided to change substantially the opening of the MS. by focusing stright away on workaholism. We hope that in this way the introductory part of the MS. has become more effective and, at the same time, to the point.
- 1
“In recent years, clinical, organizational, and occupational health scholars have paid increasing attention towards the phenomenon of workaholism – an individual characteristic which mainly manifests through working for very long hours, well-beyond what is reasonably expected of the individual [1–3]. Specifically, …”
Please include an example to illustrate the questionnable association between job performance and workaholism (lines 47-49).
As requested, we explained why workaholism may not help job performance:
- 1
“Additionally, it has been questioned that workaholics may perform better than their colleagues, since they have perfectionistic tendencies, including rigidity, inflexibility, and difficulties in delegating tasks [16,17], which do not help job performance. Indeed, a recent study found that workaholism was not associated with a supervisor-related measure of job performance [11], suggesting that there may be little advantage of being a workaholic for both individuals and organizations.”
Can you please briefly define the concept « hot temperament » (line 66)?
As suggested, we explained that it “characterizes individuals who tend to handle stress by reacting aggressively” (p. 2). We took this notion from the seminal work by Anderson and Pearson (1999) and the paper on workplace aggression on Annual Review of Psychology by Barling et al. (2009). We quote both papers in the references section.
The Coping strategy of venting is more nuanced, as other authors have also found positive effects according to the situation (line 77).
For example:
Smith MM, Saklofske DH, Keefer KV, Tremblay PF. Coping Strategies and Psychological Outcomes: The Moderating Effects of Personal Resiliency. J Psychol. 2016;150(3):318-32. doi: 10.1080/00223980.2015.1036828. Epub 2015 May 7. PMID: 25951375.
Dunkley DM, Lewkowski M, Lee IA, Preacher KJ, Zuroff DC, Berg JL, Foley JE, Myhr G, Westreich R. Daily Stress, Coping, and Negative and Positive Affect in Depression: Complex Trigger and Maintenance Patterns. Behav Ther. 2017 May;48(3):349-365. doi: 10.1016/j.beth.2016.06.001. Epub 2016 Jun 9. PMID: 28390498.
We agree with this. In Smith et al (2016), the negative effect of “emotion-focused coping” on depression, anxiety, etc., was moderated by personal resilience. Thus, we are not sure this is very relevant to our study. In Dunkley et al. (2017), the avoidant coping factor was indexed by behavioral disengagement, mental disengagement, and denial (see p. 356). Therefore, also in this case, we are not sure this study may be of help for strengthening our MS. However, inspired by your suggestions, we now report another study (Krischer et al., 2010) showing that venting emotion via CWB was beneficial for the invidual level of emotional exhaustion. The idea is that, by venting emotions through CWB, the individual may reduce the experienced level of stress and tension. We have added in the MS. the following sentence:
- 2
“Although discharging negative emotions may be beneficial for the individual (Krischer et al., 2010), it may come with significant interpersonal costs if it takes the form of inappropriate behavior”
Materials and methods: Can you please specify the time gap between T1 and T2 in this section?
The time lag was reported in the opening of the “Participants and procedure” section (p. 3):
“Data were collected in a two-wave, one-year lagged study in a national healthcare service organization in Northern Italy.”
As mentioned before, the sample consisting of 86% female nurses may represent a bias in the study, as well as all from the same workplace setting. How do you address this issue?
Statistically, we controlled for gender in the main analyses to partial out the variance explained by gender in the main criterion variable (i.e., enactment of bullying). Despite this, we agree with you that the characteristics of the sample may substantially limit the generalizability of our results. Thus, we strengthened the limitations section reported in the final part of the MS. in the following way:
- 10
“Third, our sample was relatively small, made in large part by females nurses, and involved individuals employed in the same public organization, resulting in a limited generalizability to other organizational contexts (e.g., industry), jobs and sectors.”.
We would like to insist, however, that the relationship found in our study between workaholism and enactment of bullying is original and theoretically plausible, thus contributing to the literature on the two phenomena. Of course, we agree that replication is absolutely necessary.
In the measures section, please specify the direction of the significant correlations (line 129).
Done, thank you.
- 4
“The Dutch Work Addiction Scale has been adapted in Italian, showing good internal consistency and test-retest stability. It also showed significant negative correlations with measures of well-being and positive correlations with measures of poor mental health such as anxiety and depressive symptoms [46].”
The three scales used (Dutch Work Addiction Scale, Job-related Affective Well-being Scale and the Short Negative Acts Questionnaire) use 1-4, 1-5 and 1-5 Likert scales respectively. How did you address this difference in the analyses? Transforming the scores into Z-scores is sufficient?
We adopted the validated versions of the three scales, including the original response format. It may be risky to alter the psychometrics of a measurement tool – for example, its reliability can change. Having scales with different response formats is frequent. Take for example one of the studies you suggested above (Smith et al., 2016), published on a good outlet such as Journal of Psychology. In that study, the authors used a 5-point response format to measure coping strategies, and a 3-point response format to measure anxiety, depression, and stress. Just to give you another example from a study on workaholism which we included in our reference list (i.e., Shimazu et al., 2010), the authors used the DUWAS with 1-4 response scale and used a 1-10 single item measure to assess job performance. In our case, we only have one scale (the DUWAS) with a one point less in its response format as compared to the other scales. Honestly, although we acknowledge this might be an overall problem for the field, we do not feel this is an issue that mertis further elaboration in our MS. However, if you do have strong reasons to believe that this may have affected our results, we will include the different response format of the adopted scales as a further limitation of our study.
The use of the Short Negative Acts Questionnaire with a perpetrator’s perspective is not convincing, given the alpha is quite low (.70). What did the other mentioned studies find as alphas? Please specify. Perhaps the scale presents a flaw in some of the items and their formulations, not to mention the social desirability issue for measuring bullying from the perpetrator’s perspective?
We agree that the adopted measure of enacted bullying may have been affected by social desirability, as all measures of CWB, and as we already highlighted in the first version of our MS:
- 10
“… self-reports of the enactment of aggressive behavior are likely biased by social desirability, implying that further studies employing alternative methods, such as supervisors and co-workers’ reports of workplace bullying, are needed for reaching more solid conclusions about the relationship between workaholism and bullying”
To reduce the impact of such bias we emphasised with participants that the questionnaire was anonimous and that the health care organization was only interested in aggregate scores at the group level. Baillien et al (2011) used exactly the same scale of enactment of bullying and obtained similar internal consistency, that is .66 at time 1 and .68 at time 2. They also obtained means and standard deviations (T1=1.30[SD=0.26], T2=1.29[SD=0.26]) similar to urs (T1=1.31[SD=0.30], T2=1.20[SD=0.31]). In a similar study using a different tool, namely the abuse/hostility subscale of the CWB-checklist (Spector et al., 2006 - https://doi.org/10.1016/j.jvb.2005.10.005), Balducci and colleagues (2011) obtained an alpha of .71; however, the measure consisted of 12 items (i.e., 3 more than the Short Negative Acts Questionnaire used here).
It is also possible that for behavioral phenomena such as enactment of bullying, items are formative (rather than reflective) indicators (see Spector et al., 2006), suggesting that internal consistency may not be totally appropriate as an indicator of reliability. Acknowldging your comment, we have now reported a further limitation in the final part of the MS. related to the suboptimal internal consistency of the bullying behavior measure. About this, please see also the additional explanations in response to your further comment on the bullying measure below.
- 10
“Additionally, the adopted measure of bullying obtained a suboptimal internal consistency, meaning that measurement error for the construct may have been substantial, potentially affecting the obtained results. In additional analyses (available upon request from the first author) we found that one of the items of the measure (i.e., “I carried out practical jokes on someone I don’t get on with”) had a very low loading at one measurement occasion on the underlying factor (i.e. < .10). However, discarding the item did not significantly increase the internal consistency of the measure (alpha was .68 and became .69). Future studies should pay attention to this problematic item when using the SNAQ as a measure of enacted bullying, and perhaps consider deleting or reformulating it. It should also be noted, however, that according to some scholars [52] behavioral items of aggression such as the one adopted here may be considered ‘formative’ indicators, for which internal consistency and factor analaysis may not be appropriate to assess their structure and reliability.”
In the data analysis section, how does the gender covariate in model 1 correct for or take into account the unequal proportion of males (14%) and females (86%) in the study?
This is explained in the data analysis section, where we reported that “Model 1 also included gender as a covariate affecting T2 enactment of bullying behavior, since research has often shown that males tend to be more aggressive than females (e.g., [23])”. By incuding gender, we partialled out the variance in enactment of bullying that was explained by such sociodemographic variable. We hope this is now clearer.
Results: Results show that T1 workaholism was a risk factor for drop out. However, it is argued that this "drop-out issue did not biais the results because selection bias only occurs if drop-out from the study is related to both predictor(s) and the outcome variable". Do you have more references to support this statement or more explanation for this please?
We have consulted “Occupational and Environmental Health” by Levy et al. (eds.), Oxford University Press (2011, sixth edition):
On p. 521, it is reported that “To assess whether they [i.e., participants] are different than participants who remained in the study, participants who are lost to follow-up or refuse to participate in the study should be evaluated in terms of their potential exposure and health outcome information. Selection of study participants on either exposure or disease alone may not result in selection bias. For example, an investigation may be limited to workers with only high or low exposure, rather than on all workers in a industry or plant. For selection bias to be a potential problem, inclusion in the study must be related to both exposure and health outcomes”.
In any case, we have tempered a bit our prior sentence, since we believe that a range restriction in one of the crucial study variables may potentially be consequential. Now we report that:
- 6
“Although these results may have determined a range restriction in workaholism with potential implications for the main analyses, it is also true that a selection bias is thought to be serious concern if drop‐out from the study is related to both the predictor(s) and the outcome variable [58], which was not the case in the present study.”
In table 1, the p values are missing in the notes below. Please specify whether it’s p<0.05 or p<0.01. Moreover, the significant correlations should be identifiable accordingly in the table with * or **.
The scores show quite low mean ranges for bullying, which can again indicate problematic items or measurement issues with this scale. Please do a CFA for each scale to verify which items should be re-formulated in future studies.
As suggested, we have now included asterisks clarifying the level of significance of each correlation (see new Table 1).
The low mean values of enacted bullying are a usual phenomenon in research in this area, as in CWB research (e.g., see above the mean values obtained in Baillien et al. in a similar study on enacted bullying). As requested, we report below the fit statistics for a 1-factor solution (MLR estimation) of the NAQ at both T1 and T2. Note, however, that we did not expect a very good fit for this measure. For instance, Spector et al (2006 - https://doi.org/10.1016/j.jvb.2005.10.005), in developing their scale of CWB, contended that items such as these can be thougth of as formative indicators rather than reflective indicators, meaning that statistics such as internal consitency and factor analysis may not be appropriare to assess their scale properties. They report that (p. 451):
“We chose expert judgment of item content over factor analysis of items for two reasons. First, item checklists such as our CWB-C are causal indicator scales for which items are not interchangeable measures of a single underlying construct (Bollen & Lennox, 1991; Edwards & Bagozzi, 2000). Such scales often are comprised of items that are not highly related and thus don’t form the sorts of factors that the more typical effect indicator scales produce. Second, the items ask respondents to report frequency of engaging in each behavior. As seen in Table 3, there was considerable variability in the percentage of people who engaged in each behavior, with many of the items rarely endorsed. This produced differential skew in the distribution of many items, and differences in underlying distribution shape can cause distortions in factor structures (e.g., Spector, Van Katwyk, Brannick, & Chen, 1997)”.
Spector et al (2006) did not even compute internal consistency for their measure of CWB.
However, in some other measures of CWB, such as the Bennet and Robinson (2000)’s workplace deviance scale (see https://www.researchgate.net/publication/304188906_Workplace_Deviance) internal consistency was reported.
Considering these arguments, in additional analyses (currently not reported in paper) we carried out a CFA on the SNAQ and, by including covariation of a couple of error variances (the same at both T1 and T2), we found acceptable solutions. The fit was the following at T1 (χ2(25) = 42.83, CFI = .89; RMSEA = .055; SRMR = .056), while it was the following at T2 (χ2(25) = 44.79, CFI = .91; RMSEA = .058; SRMR = .048). Inspection of factor loadings revealed that indeed one item (“I carried out practical jokes on someone I don’t get on with”) had a low factor loading at T1 (<.10), with a slightly higher value estimated at T2 (.17). However, removing this item at both times from the scale did not substantially improve the internal consistency (at T1 alpha was .68 and became .69, at T2 alpha remained .65), meaning that measurement error remained more or less the same. Thus, we decided to keep the measure as it is, while acknowledging your concerns by including the low loading of the item among the limitations of the paper. As recommended, we also suggested that future studies should pay attention to this problematic item when using the SNAQ as a measure of enacted bullying, and perhaps consider to discard or reformulate it. Thanks a lot for this very useful comment.
In Figure 1, H1 shows a rather small effect of .10* between T1 workaholism and T2 Enactment of workplace bullying behavior. Perhaps, this initial weak relationship may explain why only a tendencial effect (.13, t(1). = 1.78, p = .08) was found in H2 for the mediating effect of anger. Again, a more thorough examination at the items level would be helpful for all three scales.
Thank you for yout toughtful comment. See our response to your previous comment on the adopted bullying measure. We have included more elaborate limitations now in the final section of the MS.
For future studies, the use of daily or weekly job-related diaries is a great suggestion. I suggest to rephrase the sentence in lines 332-337 because it is not clear. I also suggest using a positive formulation to ease the comprehension.
As suggested, we better clarified that sentence by using a positive formulation to ease the comprehension:
- 10
“In conclusion, our study adds to the previous literature by showing that workaholism may be a potential antecedent of workplace aggressive behavior. The main implication of our results is that employers and practitioners have one additional reason for planning and conducting interventions aimed at reducing workaholism in the workforce, that is to prevent the enactment of CWB in the form of bullying, with negative consequences on employees and the organization overall.”
Dear Reviewer, once again thank you for your careful reading of our MS. and for your constructive comments. They required quite some work. We hope that we have been able to improve our MS., making it adequate for publication on IJERPH.
Reviewer 2 Report
General: This article is quite original in its approach to studying workaholism, bullying (from the the perpetrator's perspective) and the role of anger. It's an overall very interesting article. The longitudinal perspective and the inclusion of anger as a potential mediator in the relationship between workaholism and bullying is a novelty and merits to be pursued in the future. However, the design of the study has some flaws. For instance, the effect found for H1 seems quite small, most probably due to measurement issues. Moreover, the sample represents an overly large proportion of females (86%), which may have biased the results. Also, the dropout rate seems problematic from T1 to T2 and may also explain why the relationship between workaholism and bullying is so small. Recommendations on how to improve future longitudinal studies to examine the relationships between workaholism, bullying at work and anger give some good indications. Several more suggestions are described for each section.
Abstract: It's not clear whether anger mediates the relationship or not. The sentence summarizing the effect found at T1 and T2 (lines 21-23) is not very clear.
Introduction: The first two sentences are a little too broad and generic. The introduction may benefit from stronger argumentation and a few specific numbers perhaps demonstrating the changes at work(lines 30-33). Which Western countries do you include here?
Please include an example to illustrate the questionnable association between job performance and workaholism (lines 47-49).
Can you please briefly define the concept « hot temperament » (line 66)?
The Coping strategy of venting is more nuanced, as other authors have also found positive effects according to the situation (line 77).
For example:
Smith MM, Saklofske DH, Keefer KV, Tremblay PF. Coping Strategies and Psychological Outcomes: The Moderating Effects of Personal Resiliency. J Psychol. 2016;150(3):318-32. doi: 10.1080/00223980.2015.1036828. Epub 2015 May 7. PMID: 25951375.
Dunkley DM, Lewkowski M, Lee IA, Preacher KJ, Zuroff DC, Berg JL, Foley JE, Myhr G, Westreich R. Daily Stress, Coping, and Negative and Positive Affect in Depression: Complex Trigger and Maintenance Patterns. Behav Ther. 2017 May;48(3):349-365. doi: 10.1016/j.beth.2016.06.001. Epub 2016 Jun 9. PMID: 28390498.
Materials and methods: Can you please specify the time gap between T1 and T2 in this section? As mentioned before, the sample consisting of 86% female nurses may represent a bias in the study, as well as all from the same workplace setting. How do you address this issue?
In the measures section, please specify the direction of the significant correlations (line 129).
The three scales used (Dutch Work Addiction Scale, Job-related Affective Well-being Scale and the Short Negative Acts Questionnaire) use 1-4, 1-5 and 1-5 Likert scales respectively. How did you address this difference in the analyses? Transforming the scores into Z-scores is sufficient?
The use of the Short Negative Acts Questionnaire with a perpetrator’s perspective is not convincing, given the alpha is quite low (.70). What did the other mentioned studies find as alphas? Please specify. Perhaps the scale presents a flaw in some of the items and their formulations, not to mention the social desirability issue for measuring bullying from the perpetrator’s perspective?
In the data analysis section, how does the gender covariate in model 1 correct for or take into account the unequal proportion of males (14%) and females (86%) in the study?
Results: Results show that T1 workaholism was a risk factor for drop out. However, it is argued that this "drop-out issue did not biais the results because selection bias only occurs if drop-out from the study is related to both predictor(s) and the outcome variable". Do you have more references to support this statement or more explanation for this please?
In table 1, the p values are missing in the notes below. Please specify whether it’s p<0.05 or p<0.01. Moreover, the significant correlations should be identifiable accordingly in the table with * or **.
The scores show quite low mean ranges for bullying, which can again indicate problematic items or measurement issues with this scale. Please do a CFA for each scale to verify which items should be re-formulated in future studies.
In Figure 1, H1 shows a rather small effect of .10* between T1 workaholism and T2 Enactment of workplace bullying behavior. Perhaps, this initial weak relationship may explain why only a tendencial effect (.13, t(1). = 1.78, p = .08) was found in H2 for the mediating effect of anger. Again, a more thorough examination at the items level would be helpful for all three scales.
For future studies, the use of daily or weekly job-related diaries is a great suggestion. I suggest to rephrase the sentence in lines 332-337 because it is not clear. I also suggest using a positive formulation to ease the comprehension.
Author Response
See attached file

Dear Editor and Reviewers,
thank you for giving us the possibility to submit a revised version of our manuscript (MS.) on workaholism and the enactment of bullying behavior. We have appreciated very much the constructive tone of both reviews. We respond below point-by-point to each Reviewer comment – see text in red colour. Addressing all the comments required quite much work. We hope that we have been able to strengthen our MS., making it suitabile for publication on IJERPH. However, if you have any remaining/additional comment and suggestion, we will be happy to address them.
Changes in the manuscript are highlighted with red and underlined font.
Best regards,
Cristian Balducci,
On behalf of all the authors.
Reviewer 1 comments
Although the paper seems to focus on an important topic, specific problems in this manuscript deserve attention. I present them here to make some ideas for strengthening the article.
Dear Reviewer,
first of all thank you very much for reading and commenting our study. Thank you also for being constructive and giving specific suggestions on how to strengthen our manuscript (MS.). We reply here to your fist block of comments regarding the Introduction of the MS.
Changes in the manuscript are highlighted with red and underlined font.
The authors should focus more on addressing what we already know about the topic before bringing in a gap considering what the paper tries to fill in. This would make it clear to the reader why it is crucial to address the shortcomings in the literature. I believe that your contribution would be more significant if you present convincing arguments regarding why there is a need to investigate the underlying questions. What new insight is your study offering to readers? If you convince the reader of the necessity for your work to expand our current knowledge in the introduction section, you would significantly enhance your contribution.
Please add the theoretical perspective to explain the association and build arguments among study variables.
I suggest reviewing the past studies, identifying all the predictors and outcomes of workaholism and workplace bullying, and putting them in a table with relevant sources in the literature review to understand both the phenomenas better. You could introduce a pre-version of this table in the introduction in the form of a paragraph. With the help of this one, one can easily understand what we know and what we do not know.
Why job-related negative affect is best to explain the association between workaholism and workplace bullying. More justifications are required with the help of literature that explored the mediating role of job-related negative affect.
Following your comments above, we have substantially extended the Introduction (pp. 1-6). We have now introduced a paragraph on the theoretical frameworks that can be used to explain the relationhip between workaholism and the enactment of bullying behavior. We opted for an established model - the stressor emotion model of counterproductive work behavior proposed by Spector and Fox (2005), which builds upon the frustration-aggression hypothesis as revisited by Berkowitz (e.g., 1989). Such a framework is particularly suited to shed light on the relationship between workaholism and enactment of bullying, since it postulates that aggression (i.e., bullying) may be a consequence of negative affective states (frustration and tension) experienced in relation to external stressors (e.g., negative working conditions) or personal dispositions (‘internal’ factors) fueling the experience of stress. We also mention an additional theoretical model developed in social psychology, namely the General Affective Aggression Model (Allen et al., 2018 – see reference list in the MS.), which makes predictions very similar to the stressor-emotion model, that is, aggression may be a consequence of individual differences that fuel critical internal states such as negative affect.
As we now explain by describing in more detail the results emerged from recent sound studies (e.g., the two studies by Balducci et al., 2021), workaholism may fuel work-related stress and expose the individual to frustration, tension, and negative affect, creating the necessary and sufficient internal experiences that trigger the enactment of bullying behavior. In this way, we have provided the requested theoretical rationale for our study, also explaining “why job-related negative affect is best to explain the association between workaholism and workplace bullying”. On the latter point, we have also included some recent sound studies indicating that workaholics indeed experience stress and negative job-related affect, making it plausible that negative affective experiences are a crucial mediator of the workaholism-enactment of bullying relationship.
In the opening section of the Introduction (p. 3), we also provided additional reasons, as per your request above, to investigate the workaholism-enactment of bullying relationship. We have now emphasised that workaholics often have responsibility positions, so if we demonstrate that workaholism is related to the enactment of bullying we will provide important evidence that workaholics may not be the best fit for managerial positions, since they contribute to create toxic workplaces.
We would avoid, however, to extend too much the Introduction with the addition of more text or tables regarding both workaholism and workplace bullying, which are well known phenomena and for which we have now inclued additional key references, also considering that the authors’ guidelines of IJERPH stress that the Intro should be brief and to the point: “The introduction should briefly place the study in a broad context and highlight why it is important”.
In sum, we believe that the revised version of our manuscript fits well with IJERPH guidelines while, at the same time, responding to the request of identifying the core results of the past literature and providing a more comprehensive theoretical rationale for our study.
-Can you describe how you recruited the sample? Participants and procedures should need to elaborate in more detail.
We have given additional information for the section “Participants and procedure” (p. 3), hoping that this will provide sufficient details on the recruitment procedures. Alternatively, please let us know which specific point is in need of further clarification.
-Please justify why MPlus software is best to investigate the study paths instead of other techniques like SEM.
We believe this comment raised from a misunderstanding of the difference between Mplus (i.e., a widely used data analysis software focused on SEM, similar to other softwares such as Lisrel or the R package lavaan) and SEM (i.e., a class of models that use the covariance matrix for estimating target model parameters, and which can be implemented with the Mplus software). Path analysis is just a specific type of SEM that uses observed variables only (i.e., aggregate scales scores, with no latent variables and factor loadings), and which is routinely used to conduct multivariate analyses, especially when the sample size is not sufficiently large to estimate all the parameters involved in SEM with latent variables. Thus, we did not find necessary to justify the use of Mplus to investigate the study path, since both the Mplus software and SEM with observed variables are accepted and widely used by the scientific community. However, to account for possible misunderstandings, we now referenced (p. 5) to Tabachnick and Fidell (2013). Using multivariate statistics, Boston: Pearson Education.
-The convergent and discriminant validity needs to be performed before testing the direct and indirect paths.
Some results are already evident from the correlation table (see Table 1), showing that, in all cases, the measures of the same construct at different times (T1 and T2) correlate more strongly than the measures of different constructs.
Additionally, we conducted two confirmatory factor analyses (estimator: robust RML), one at T1 and one at T2 to test for whether the three constructs under investigation (i.e, workaholism, enactment of bullying, and job-related negative affect) could be discriminated empirically. In these analyses, also to account for the relatively low sample size, each construct was operationalized by means of two parcels (i.e., subscales). For example, workaholism was operationalized by two 5-item parcels, one reflecting working excessively, and the other reflecting working compulsively. At T1, a 3-factor model fitted the data sufficiently well (χ2(6) = 13.49, CFI = .96; RMSEA = .073; SRMR = .030) and significantly better than a 1-factor model (Δχ2(3) = 76.44, p<.001). At T2 the results were similar, with the 3-factor model (χ2(6) = 5.72, CFI = 1.00; RMSEA = .000; SRMR = .027) fitting significantly better than the 1-factor model (Δχ2(3) = 119.15, p<.001).
Currently, we did not include such results in the Results section since the three considered measures are well validated and have been employed multiple times in previous research, and our study did not focus on construct validity. However, if you believe we should integrate the above results in the MS., we will be happy to do so.
-The discussion part is underdeveloped. In particular, the discussion needs to be more thorough and linked to the literature review.
-The theoretical implications part is underdeveloped.
As suggested, we substantially expanded the Discussion (pp. 8-9), particularly by better linking our results to the most relevant literature, and by better elaborating on the implications of our results. We also acknowledged additional limitations in reply to the Reviewer 2. In all, there are three additional paragraphs in the Discussion. We believe the current version of the Discussion is now well developed, providing the necessary information to interpret our results, to understand the study limitations, and to appreciate both the practical and theoretical implications of our study.
I am very disappointed that most of the references are very old. The more references that come from the past five years, the better. Therefore, I suggest that the authors need to incorporate recent and more context-related articles related to the phenomenon.
We have now added more recent references, also in response to the other comments of both reviewers. However, we highlight that thirty out of fifty-five references (i.e. the majority) reported in the first version of the MS. were published in 2016 or later, so we kindly disagree with your statement that “most references are very old”.
In general, I would like my recommendations to help the authors improve their work. I hope the authors will benefit from these suggestions and make the necessary amendments to improve the work for later submission.
Dear Reviewer, thank you again for your recommendations, and we hope that, by following your and Reviewer 2’s suggestions, we have been able to improve our MS. to the point of being accepted for publication on IJERPH.
Reviewer 2 comments
General: This article is quite original in its approach to studying workaholism, bullying (from the the perpetrator's perspective) and the role of anger. It's an overall very interesting article. The longitudinal perspective and the inclusion of anger as a potential mediator in the relationship between workaholism and bullying is a novelty and merits to be pursued in the future. However, the design of the study has some flaws. For instance, the effect found for H1 seems quite small, most probably due to measurement issues. Moreover, the sample represents an overly large proportion of females (86%), which may have biased the results. Also, the dropout rate seems problematic from T1 to T2 and may also explain why the relationship between workaholism and bullying is so small. Recommendations on how to improve future longitudinal studies to examine the relationships between workaholism, bullying at work and anger give some good indications. Several more suggestions are described for each section.
Dear Reviewer 2, thank you very much for appreciating our study and for your comments and suggestions on how to improve it. We have now acknowledged the limitations you report above in a more elaborated way (p. 10), also giving suggestions on how future studies could overcome (some of) them.
Abstract: It's not clear whether anger mediates the relationship or not. The sentence summarizing the effect found at T1 and T2 (lines 21-23) is not very clear.
Thank you for noting this. We agree with you and have now better clarified that part:
- 1
“To shed light on the role of a potential mechanism explaining the link between workaholism and enactment of bullying we examined whether job-related negative affect (e.g., anger) mediated their longitudinal relationship. However, whereas increased negative affect from T1 to T2 was positively associated with T2 enacted workplace bullying, the relationship between T1 workaholism and increased job-related negative affect was not significant, contrary to the hypothesized mediation.”
Introduction: The first two sentences are a little too broad and generic. The introduction may benefit from stronger argumentation and a few specific numbers perhaps demonstrating the changes at work (lines 30-33). Which Western countries do you include here?
We agree also on this. However, we decided to change substantially the opening of the MS. by focusing stright away on workaholism. We hope that in this way the introductory part of the MS. has become more effective and, at the same time, to the point.
- 1
“In recent years, clinical, organizational, and occupational health scholars have paid increasing attention towards the phenomenon of workaholism – an individual characteristic which mainly manifests through working for very long hours, well-beyond what is reasonably expected of the individual [1–3]. Specifically, …”
Please include an example to illustrate the questionnable association between job performance and workaholism (lines 47-49).
As requested, we explained why workaholism may not help job performance:
- 1
“Additionally, it has been questioned that workaholics may perform better than their colleagues, since they have perfectionistic tendencies, including rigidity, inflexibility, and difficulties in delegating tasks [16,17], which do not help job performance. Indeed, a recent study found that workaholism was not associated with a supervisor-related measure of job performance [11], suggesting that there may be little advantage of being a workaholic for both individuals and organizations.”
Can you please briefly define the concept « hot temperament » (line 66)?
As suggested, we explained that it “characterizes individuals who tend to handle stress by reacting aggressively” (p. 2). We took this notion from the seminal work by Anderson and Pearson (1999) and the paper on workplace aggression on Annual Review of Psychology by Barling et al. (2009). We quote both papers in the references section.
The Coping strategy of venting is more nuanced, as other authors have also found positive effects according to the situation (line 77).
For example:
Smith MM, Saklofske DH, Keefer KV, Tremblay PF. Coping Strategies and Psychological Outcomes: The Moderating Effects of Personal Resiliency. J Psychol. 2016;150(3):318-32. doi: 10.1080/00223980.2015.1036828. Epub 2015 May 7. PMID: 25951375.
Dunkley DM, Lewkowski M, Lee IA, Preacher KJ, Zuroff DC, Berg JL, Foley JE, Myhr G, Westreich R. Daily Stress, Coping, and Negative and Positive Affect in Depression: Complex Trigger and Maintenance Patterns. Behav Ther. 2017 May;48(3):349-365. doi: 10.1016/j.beth.2016.06.001. Epub 2016 Jun 9. PMID: 28390498.
We agree with this. In Smith et al (2016), the negative effect of “emotion-focused coping” on depression, anxiety, etc., was moderated by personal resilience. Thus, we are not sure this is very relevant to our study. In Dunkley et al. (2017), the avoidant coping factor was indexed by behavioral disengagement, mental disengagement, and denial (see p. 356). Therefore, also in this case, we are not sure this study may be of help for strengthening our MS. However, inspired by your suggestions, we now report another study (Krischer et al., 2010) showing that venting emotion via CWB was beneficial for the invidual level of emotional exhaustion. The idea is that, by venting emotions through CWB, the individual may reduce the experienced level of stress and tension. We have added in the MS. the following sentence:
- 2
“Although discharging negative emotions may be beneficial for the individual (Krischer et al., 2010), it may come with significant interpersonal costs if it takes the form of inappropriate behavior”
Materials and methods: Can you please specify the time gap between T1 and T2 in this section?
The time lag was reported in the opening of the “Participants and procedure” section (p. 3):
“Data were collected in a two-wave, one-year lagged study in a national healthcare service organization in Northern Italy.”
As mentioned before, the sample consisting of 86% female nurses may represent a bias in the study, as well as all from the same workplace setting. How do you address this issue?
Statistically, we controlled for gender in the main analyses to partial out the variance explained by gender in the main criterion variable (i.e., enactment of bullying). Despite this, we agree with you that the characteristics of the sample may substantially limit the generalizability of our results. Thus, we strengthened the limitations section reported in the final part of the MS. in the following way:
- 10
“Third, our sample was relatively small, made in large part by females nurses, and involved individuals employed in the same public organization, resulting in a limited generalizability to other organizational contexts (e.g., industry), jobs and sectors.”.
We would like to insist, however, that the relationship found in our study between workaholism and enactment of bullying is original and theoretically plausible, thus contributing to the literature on the two phenomena. Of course, we agree that replication is absolutely necessary.
In the measures section, please specify the direction of the significant correlations (line 129).
Done, thank you.
- 4
“The Dutch Work Addiction Scale has been adapted in Italian, showing good internal consistency and test-retest stability. It also showed significant negative correlations with measures of well-being and positive correlations with measures of poor mental health such as anxiety and depressive symptoms [46].”
The three scales used (Dutch Work Addiction Scale, Job-related Affective Well-being Scale and the Short Negative Acts Questionnaire) use 1-4, 1-5 and 1-5 Likert scales respectively. How did you address this difference in the analyses? Transforming the scores into Z-scores is sufficient?
We adopted the validated versions of the three scales, including the original response format. It may be risky to alter the psychometrics of a measurement tool – for example, its reliability can change. Having scales with different response formats is frequent. Take for example one of the studies you suggested above (Smith et al., 2016), published on a good outlet such as Journal of Psychology. In that study, the authors used a 5-point response format to measure coping strategies, and a 3-point response format to measure anxiety, depression, and stress. Just to give you another example from a study on workaholism which we included in our reference list (i.e., Shimazu et al., 2010), the authors used the DUWAS with 1-4 response scale and used a 1-10 single item measure to assess job performance. In our case, we only have one scale (the DUWAS) with a one point less in its response format as compared to the other scales. Honestly, although we acknowledge this might be an overall problem for the field, we do not feel this is an issue that mertis further elaboration in our MS. However, if you do have strong reasons to believe that this may have affected our results, we will include the different response format of the adopted scales as a further limitation of our study.
The use of the Short Negative Acts Questionnaire with a perpetrator’s perspective is not convincing, given the alpha is quite low (.70). What did the other mentioned studies find as alphas? Please specify. Perhaps the scale presents a flaw in some of the items and their formulations, not to mention the social desirability issue for measuring bullying from the perpetrator’s perspective?
We agree that the adopted measure of enacted bullying may have been affected by social desirability, as all measures of CWB, and as we already highlighted in the first version of our MS:
- 10
“… self-reports of the enactment of aggressive behavior are likely biased by social desirability, implying that further studies employing alternative methods, such as supervisors and co-workers’ reports of workplace bullying, are needed for reaching more solid conclusions about the relationship between workaholism and bullying”
To reduce the impact of such bias we emphasised with participants that the questionnaire was anonimous and that the health care organization was only interested in aggregate scores at the group level. Baillien et al (2011) used exactly the same scale of enactment of bullying and obtained similar internal consistency, that is .66 at time 1 and .68 at time 2. They also obtained means and standard deviations (T1=1.30[SD=0.26], T2=1.29[SD=0.26]) similar to urs (T1=1.31[SD=0.30], T2=1.20[SD=0.31]). In a similar study using a different tool, namely the abuse/hostility subscale of the CWB-checklist (Spector et al., 2006 - https://doi.org/10.1016/j.jvb.2005.10.005), Balducci and colleagues (2011) obtained an alpha of .71; however, the measure consisted of 12 items (i.e., 3 more than the Short Negative Acts Questionnaire used here).
It is also possible that for behavioral phenomena such as enactment of bullying, items are formative (rather than reflective) indicators (see Spector et al., 2006), suggesting that internal consistency may not be totally appropriate as an indicator of reliability. Acknowldging your comment, we have now reported a further limitation in the final part of the MS. related to the suboptimal internal consistency of the bullying behavior measure. About this, please see also the additional explanations in response to your further comment on the bullying measure below.
- 10
“Additionally, the adopted measure of bullying obtained a suboptimal internal consistency, meaning that measurement error for the construct may have been substantial, potentially affecting the obtained results. In additional analyses (available upon request from the first author) we found that one of the items of the measure (i.e., “I carried out practical jokes on someone I don’t get on with”) had a very low loading at one measurement occasion on the underlying factor (i.e. < .10). However, discarding the item did not significantly increase the internal consistency of the measure (alpha was .68 and became .69). Future studies should pay attention to this problematic item when using the SNAQ as a measure of enacted bullying, and perhaps consider deleting or reformulating it. It should also be noted, however, that according to some scholars [52] behavioral items of aggression such as the one adopted here may be considered ‘formative’ indicators, for which internal consistency and factor analaysis may not be appropriate to assess their structure and reliability.”
In the data analysis section, how does the gender covariate in model 1 correct for or take into account the unequal proportion of males (14%) and females (86%) in the study?
This is explained in the data analysis section, where we reported that “Model 1 also included gender as a covariate affecting T2 enactment of bullying behavior, since research has often shown that males tend to be more aggressive than females (e.g., [23])”. By incuding gender, we partialled out the variance in enactment of bullying that was explained by such sociodemographic variable. We hope this is now clearer.
Results: Results show that T1 workaholism was a risk factor for drop out. However, it is argued that this "drop-out issue did not biais the results because selection bias only occurs if drop-out from the study is related to both predictor(s) and the outcome variable". Do you have more references to support this statement or more explanation for this please?
We have consulted “Occupational and Environmental Health” by Levy et al. (eds.), Oxford University Press (2011, sixth edition):
On p. 521, it is reported that “To assess whether they [i.e., participants] are different than participants who remained in the study, participants who are lost to follow-up or refuse to participate in the study should be evaluated in terms of their potential exposure and health outcome information. Selection of study participants on either exposure or disease alone may not result in selection bias. For example, an investigation may be limited to workers with only high or low exposure, rather than on all workers in a industry or plant. For selection bias to be a potential problem, inclusion in the study must be related to both exposure and health outcomes”.
In any case, we have tempered a bit our prior sentence, since we believe that a range restriction in one of the crucial study variables may potentially be consequential. Now we report that:
- 6
“Although these results may have determined a range restriction in workaholism with potential implications for the main analyses, it is also true that a selection bias is thought to be serious concern if drop‐out from the study is related to both the predictor(s) and the outcome variable [58], which was not the case in the present study.”
In table 1, the p values are missing in the notes below. Please specify whether it’s p<0.05 or p<0.01. Moreover, the significant correlations should be identifiable accordingly in the table with * or **.
The scores show quite low mean ranges for bullying, which can again indicate problematic items or measurement issues with this scale. Please do a CFA for each scale to verify which items should be re-formulated in future studies.
As suggested, we have now included asterisks clarifying the level of significance of each correlation (see new Table 1).
The low mean values of enacted bullying are a usual phenomenon in research in this area, as in CWB research (e.g., see above the mean values obtained in Baillien et al. in a similar study on enacted bullying). As requested, we report below the fit statistics for a 1-factor solution (MLR estimation) of the NAQ at both T1 and T2. Note, however, that we did not expect a very good fit for this measure. For instance, Spector et al (2006 - https://doi.org/10.1016/j.jvb.2005.10.005), in developing their scale of CWB, contended that items such as these can be thougth of as formative indicators rather than reflective indicators, meaning that statistics such as internal consitency and factor analysis may not be appropriare to assess their scale properties. They report that (p. 451):
“We chose expert judgment of item content over factor analysis of items for two reasons. First, item checklists such as our CWB-C are causal indicator scales for which items are not interchangeable measures of a single underlying construct (Bollen & Lennox, 1991; Edwards & Bagozzi, 2000). Such scales often are comprised of items that are not highly related and thus don’t form the sorts of factors that the more typical effect indicator scales produce. Second, the items ask respondents to report frequency of engaging in each behavior. As seen in Table 3, there was considerable variability in the percentage of people who engaged in each behavior, with many of the items rarely endorsed. This produced differential skew in the distribution of many items, and differences in underlying distribution shape can cause distortions in factor structures (e.g., Spector, Van Katwyk, Brannick, & Chen, 1997)”.
Spector et al (2006) did not even compute internal consistency for their measure of CWB.
However, in some other measures of CWB, such as the Bennet and Robinson (2000)’s workplace deviance scale (see https://www.researchgate.net/publication/304188906_Workplace_Deviance) internal consistency was reported.
Considering these arguments, in additional analyses (currently not reported in paper) we carried out a CFA on the SNAQ and, by including covariation of a couple of error variances (the same at both T1 and T2), we found acceptable solutions. The fit was the following at T1 (χ2(25) = 42.83, CFI = .89; RMSEA = .055; SRMR = .056), while it was the following at T2 (χ2(25) = 44.79, CFI = .91; RMSEA = .058; SRMR = .048). Inspection of factor loadings revealed that indeed one item (“I carried out practical jokes on someone I don’t get on with”) had a low factor loading at T1 (<.10), with a slightly higher value estimated at T2 (.17). However, removing this item at both times from the scale did not substantially improve the internal consistency (at T1 alpha was .68 and became .69, at T2 alpha remained .65), meaning that measurement error remained more or less the same. Thus, we decided to keep the measure as it is, while acknowledging your concerns by including the low loading of the item among the limitations of the paper. As recommended, we also suggested that future studies should pay attention to this problematic item when using the SNAQ as a measure of enacted bullying, and perhaps consider to discard or reformulate it. Thanks a lot for this very useful comment.
In Figure 1, H1 shows a rather small effect of .10* between T1 workaholism and T2 Enactment of workplace bullying behavior. Perhaps, this initial weak relationship may explain why only a tendencial effect (.13, t(1). = 1.78, p = .08) was found in H2 for the mediating effect of anger. Again, a more thorough examination at the items level would be helpful for all three scales.
Thank you for yout toughtful comment. See our response to your previous comment on the adopted bullying measure. We have included more elaborate limitations now in the final section of the MS.
For future studies, the use of daily or weekly job-related diaries is a great suggestion. I suggest to rephrase the sentence in lines 332-337 because it is not clear. I also suggest using a positive formulation to ease the comprehension.
As suggested, we better clarified that sentence by using a positive formulation to ease the comprehension:
- 10
“In conclusion, our study adds to the previous literature by showing that workaholism may be a potential antecedent of workplace aggressive behavior. The main implication of our results is that employers and practitioners have one additional reason for planning and conducting interventions aimed at reducing workaholism in the workforce, that is to prevent the enactment of CWB in the form of bullying, with negative consequences on employees and the organization overall.”
Dear Reviewer, once again thank you for your careful reading of our MS. and for your constructive comments. They required quite some work. We hope that we have been able to improve our MS., making it adequate for publication on IJERPH.
Round 2
Reviewer 1 Report
Dear Authors, I carefully re-evaluated your paper, finding it substantially improved with respect to the version. The revised version is much better organized and has higher scientific quality. Therefore, I recommended it for publication. Thank you